# A Novel Fluorescent Aptasensor Based on Real-Time Fluorescence and Strand Displacement Amplification for the Detection of Ochratoxin A

**DOI:** 10.3390/foods11162443

**Published:** 2022-08-13

**Authors:** Wei Guo, Haoyu Yang, Yunzhe Zhang, Hao Wu, Xin Lu, Jianxin Tan, Wei Zhang

**Affiliations:** 1College of Food Science and Technology, Hebei Agricultural University, Baoding 071000, China; 2Hebei Province Feed Microorganism Technology Innovation Center, Baoding 071000, China; 3Department of Physical Education, Hebei Agricultural University, Baoding 071000, China; 4College of Science and Technology, Hebei Agricultural University, Cangzhou 061100, China; 5College of Life Sciences, Hebei Agricultural University, Baoding 071000, China; 6Hebei Key Laboratory of Analysis and Control of Zoonotic Pathogenic Microorganism, Hebei Agricultural University, Baoding 071001, China

**Keywords:** ochratoxin A, strand displacement amplification (SDA), real-time fluorescence, aptasensor

## Abstract

It is urgently necessary to develop convenient, reliable, ultrasensitive and specific methods of ochratoxin A determination in food safety owing to its high toxicity. In the present study, an ultrasensitive and labeled-free fluorescent aptamer sensor combining real-time fluorescence with strand displacement amplification (SDA) was fabricated for the determination of OTA. In the presence of OTA, the OTA–aptamer combines with OTA, thus opening hairpins. Then, SDA primers specifically bind to the hairpin stem, which is used for subsequent amplification as a template. SDA amplification is initiated under the action of *Bst* DNA polymerase and nicking endonuclease. The amplified products (ssDNA) are dyed with SYBR Green II and detected with real-time fluorescence. The method has good linearity in the range of 0.01–50 ng mL^−1^, with the lowest limit of detection of 0.01 ng mL^−1^. Additionally, the fluorescent aptamer sensor shows outstanding specificity and reproducibility. Furthermore, the sensor shows excellent analytical performance in the artificial labeled detection of wheat and oat samples, with a recovery rate of 96.1~100%. The results suggest that the developed sensor has a promising potential application for the ultrasensitive detection of contaminants in food.

## 1. Introduction

Ochratoxin A (OTA) is a secondary metabolite produced by *Aspergillus* and *Penicillium* [1,2], and contaminations are widespread in all kinds of commercial products, such as nuts, cereals, corn, and crops [3]. Previous studies reported that OTA mainly damaged the kidney and liver, increasing the chances of renal cell cancer and liver cancer [4]. Meanwhile, OTA has been shown to be embryotoxic, malformed, neurotoxic, immunotoxic, genotoxic and carcinogenic [5].

It is difficult to completely remove OTA during food manufacturing and processing because OTA has high thermal stability [6]. In view of the strong toxicity of OTA, the International Agency for Research on Cancer (IARC) [7] has classified OTA as a possible carcinogen to humans (Group 2B). Many countries have formulated maximum residue levels (MRL) of OTA in food. For instance, the European Commission has set MRLs of 5 μg kg^−1^ for OTA in raw cereal grains and 2 μg kg^−1^ for grapes and grape products [8,9]; China has set 5 μg kg^−1^ for OTA in cereal products and 5 μg kg^−1^ and 10 μg kg^−1^ in grind coffee and instant coffee, respectively. However, the trace, high toxicity and complex pollution of OTA have blocked its actual detection. With the growing attention paid to human health and food safety, it is quite important to propose a fast, ultrasensitive and convenient technique for OTA determination in food products.

In order to achieve the efficient detection of OTA in complex food matrices, various detection technologies have been developed. The commonly used analytical methods include high-performance liquid chromatography (HPLC) [10], liquid chromatography tandem mass spectrometry [11] and other chromatographic methods [12,13]. Although these methods are highly specific and accurate, there are still some problems such as time-consuming sample pre-treatment and professional operators of sophisticated instruments. Relative to those detection methods, enzyme-linked immunosorbent assays (ELISA) [14] is more suitable for large-scale screening on site as it is simple to operate and specific to target. Nevertheless, the reproducibility of its results is poor, and the enzyme and antibody require more stringent storage environments. Therefore, to better understand OTA pollution and effectively reduce its harm to humans, it is urgent to establish an ultrasensitive and efficient detection method.

Due to the complexity of the matrices contaminated by OTA, it is particularly important to realize specific OTA detection in complex matrices. Aptamers are single-strand oligonucleotides (DNA or RNA) that are selected by an in vitro election process [15]. They have high specificity to the corresponding targets. Combining aptamers with SDA, the detection of mycotoxins can be converted to that of DNA. Unlike antibodies, aptamers can meet the requirements of a variety of experiments with high affinity, easy modification and high stability [16,17,18]. Taking advantage of those excellent characteristics of aptamers, various aptasensors have been built for the ultrasensitive detection of OTA through different sensing platforms such as electrochemical [19,20] and optical [21,22,23] platforms. Due to the advantages of rapidness, accuracy and high efficiency, fluorescence analysis has been widely considered to be an excellent candidate with wide application prospects [24,25]. Moreover, label-free fluorescence strategies using fluorescent dyes, such as SYBR Green II with strong interaction response emission, have attracted attention for further studies because of low cost and facile design.

To further improve the sensitivity of determination, isothermal amplification methods, including loop-mediated isothermal (LAMP) [26,27], rolling circle amplification (RCA) [28,29] and palindromic sequence amplification (PSA) [30], have been employed into fluorescent assays. Except as above, strand displacement amplification (SDA) can generate the exponential accumulation of single-stranded DNA (ssDNA) depending on the combination of strand-displacing polymerase and nicking endonuclease [31,32]. Due to its cost-effectiveness, adaptability and simplicity, SDA has been coupled with various analytical assays to construct different biosensors, such as electrochemical [33], chemiluminescent [34], and fluorescent [35], but the complex background and limited sensitivity limit its detection performance. Therefore, it is urgent to develop a convenient and reliable protocol for quantifying the levels of OTA in food.

Herein, we developed an ultrasensitive and efficient fluorescent aptasensor combining real-time fluorescence detection technology with SDA. First, we introduced an OTA aptamer with hairpin (HP). The OTA aptamer specifically recognized OTA and combined with it, leading to the opening of HP. Then the primer recognized and combined with the HP stem to initiate SDA under the action of nicking endonuclease and polymerase. A large number of ssDNA products were obtained during amplification and then dyed with free SYBR Green II. Finally, the fluorescence signals were collected and quantified with real-time fluorescence curves. The assay showed good linear relationships in the range of 0.01–50 ng mL^−1^, with the lowest limit of detection (LOD) being 0.01 ng mL^−1^. Moreover, the developed sensor was capable of detecting OTA in real food samples, including wheat and oat. The developed sensor combined the advantages of the high preciseness and sensitivity of real-time fluorescence detection and the simplicity of SDA operation. Thus, it is a promising platform for OTA detection in food samples.

## 2. Materials and Methods

### 2.1. Reagents and Apparatus

All the oligonucleotides were designed by mfold (www.unafold.org, accessed on 1 August 2022), synthesized and purified using HPLC by Sangon Biotech Co., Ltd., (Shanghai, China). The OTA-specific aptamer sequence was designed based on (5′-GATCGGGTGTGGGTGGCGTAAAGGGAGCATCGACA-3′), with a dissociation constant (Kd) of 0.2 μmol/L following Cruz et al. [36]. The dissolution of oligonucleotide sequences was carried out in TE buffer (10 mM Tris, 1 mM EDTA, pH = 8.0), and then the dissolved sequences were stored at 4 °C until use (Table 1). Deoxynucleoside triphosphate (dNTP) mixture, *Bst* DNA polymerase large fragment, Nb.BsrDI, 10×Cutsmart buffer (200 mM Tris-HAc, 500 mM KAc, 100 mM MgAc_2_, 1 g mL^−1^ BSA, pH = 7.9) and 10× ThermoPol reaction buffer (200 mM Tris-HCl, 100 mM KCl, 100 mM (NH_4_)_2_SO_4_, 20 mM MgSO_4_, 1% Triton X-100, pH = 8.8) were supplied by New England Biolabs Inc. (Beverly, MA, USA). Ochratoxin A (OTA), ochratoxin B (OTB), aflatoxin B1 (AFB1), deoxynivalenol (DON) and zearalenone (ZEA) were obtained from Pribolab (Singapore). All chemicals were of analytical grade. Ultrapure water (>18.25 MΩ) was used to prepare all solutions. The recording of the fluorescence spectra was conducted using a Step OnePlus real-time PCR machine (Applied Biosystems, New York, NY, USA). The measurement of circular dichroism (CD) spectra was carried out with a Jasco J-1500 CD spectrometer.

### 2.2. OTA Detection by the Fluorescent Aptasensor

Firstly, the OTA aptamer was heated at 95 °C for 5 min and then cooled slowly to room temperature for subsequent study. In the OTA fluorescence aptasensor, 500 nM HP, 500 nM primer, 5 mM MgSO_4_ and 50 mM NaCl and OTA (in the range of 0.01 to 1000 ng/mL) were incubated in 1× ThermoPol buffer and 1×Cutsmart buffer at 37 °C for 30 min in darkness. Then, 500 μM dNTPs, 0.8 U μL^−1^ *Bst* DNA polymerase large fragment, 1 U μL^−1^ Nb.BsrDI endonuclease and 5 × SYBR Green II were homogenized with the above mixture in a total reaction volume of 20 μL and incubated at 65 °C for 40 min. The fluorescence curve was drawn using a real-time fluorescence quantitative instrument SYBR green module. After the amplification reaction, the enzyme was inactivated at 80 °C for 20 min.

### 2.3. Specificity and Reproducibility Analysis

The toxins, including OTB, AFB1, FB1 and ZEN, were mixed together, with each toxin concentration being 100 ng mL^−1^. This toxin mixture was then mixed with and without OTA (10 ng mL^−1^). These two toxin mixtures were used to verify the specificity of the analytical approach. To evaluate the reproducibility of the method, different concentration of OTA (0.01 and 100 ng mL^−1^) were tested six times for statistical analysis.

### 2.4. Polyacrylamide Gel Electrophoresis (PAGE) Analysis

In this study, the amplification reaction products were verified with 20% PAGE. A 5 μL reaction solution mixed with 2 μL 6× loading buffer and used for the electrophoresis, which was carried out in 1× TBE buffer (89 mM Tris, 89 mM boric acid, 2 mM EDTA, pH 8.3) at 120 V for 90 min. Then, the polyacrylamide gel was dyed with 10,000× Super GelRed^®^ nucleic acid for 30 min in the dark. Gel image was acquired and analyzed by BINDA 2020D system.

### 2.5. Preparation of Real Samples and Detection

To verify the reliability and recovery of the developed fluorescence sensor in the detection of real samples, wheat and oat samples were spiked with OTA. Wheat and oat samples were crushed and passed through a 1 mm test sieve, and then, 2 mg of those samples were added into 10 mL of 80% methanol solution. The solutions were vigorously shaken for 10 min and then centrifuged at 10,000× *g* r/min for 10 min. A 0.22 μm syringe filter was used to filter the supernatants of wheat and oat. Different concentrations (1, 10 ng mL^−1^) of OTA were added into the filtrates and then the spiked food samples were mixed homogeneously and stored at 4°C as analytical samples.

## 3. Results and Discussion

### 3.1. Principle of Fluorescent Aptasensor

The principle of the OTA fluorescent aptasensor is displayed in Figure 1. In this design, the OTA aptamer was used to recognize OTA and bind with it. The random coil ssDNA structure of OTA aptamer has been reported to transform to an antiparallel G-quadruplex in the presence of OTA [37]. Thus, the binding site in the stem of HP was exposed. Afterwards, the free primers in the system specifically recognized and bonded to the binding sites, initiating the SDA and then producing large numbers of repetitive ssDNA following the addition of polymerase and nicking endonuclease. Owing to the high quantum yield of fluorescent dye SYBR Green II, it could embed in the ssDNA and confer high fluorescence. When SYBR Green II binds to ssDNA, the energy of the electrons from the ground state to the excited state is equal to that of the excitation of the light photon, so it can absorb the photon. And the electrons are excited to the excited state and then back to the lower state, thus emitting intense fluorescence. Whereas, in the absence of OTA, the stem-loop structure of HP was stable and SDA could not be initiated, resulting in the detachment of SYBR Green II dye from ssDNA and generating a negligible fluorescent signal. Moreover, the real-time fluorescence system realized the quantitative detection of OTA by monitoring the fluorescence curve during amplification.

### 3.2. The Feasibility of OTA Fluorescent Aptasensor

The rationality of the aptasensor was verified by real-time fluorescence curve and polyacrylamide gel (Figure 1). The delt F (ΔF) was the difference between the real-time fluorescence intensity (F) and the initial fluorescence intensity (F_0_) during the OTA detection process. The number of cycles for the fluorescence curve to reach the threshold line (the dash line) was defined as *Ct*. Fluorescence spectrum of reaction solution with the addition of HP, primers, polymerase and nicking enzyme in the absence/presence of OTA was further displayed in Figure 1A. Fluorescence emission spectrum was observed in the range of 13–40 amplification cycles, and the emission peak tended to flatten after 18 cycles. No fluorescence intensity was acquired in the absence of OTA, polymerase or nicking enzyme in the reaction solution (Figure 1A, curve 2–5; Figure 1B); whereas, a high fluorescence intensity was obtained when all solutions and OTA were added (Figure 1A, curve 1). The results are in accordance with expectations and reasonable. In the absence of OTA, OTA aptamer could not bind to it and the structure of HP maintained stable. Thus, the primers could not bind to HP stem and extended, generating a negligible fluorescence signal (Figure 1A, curve 2). Meanwhile, due to the lack of polymerase and/or nicking enzyme, SDA could not be performed, and no reaction products were obtained, and thus, a fluorescent signal of SYBR Green II could not be induced (Figure 1A, curve 3–5).

The reaction products were further tested using polyacrylamide gel electrophoresis (Figure 1B). In the absence of OTA, HP could not be opened and bind with the primer, which proved that false positive amplification will not occur. The PAGE results suggested the formation of antiparallel G-quadruplex, indicating that HP was opened in the presence of OTA (comparing lanes 3–5 with lane 6). In the presence of only polymerase, part of the combination of primer and template extended along the direction from 5’ to 3’-end, and the molecular weight increased slightly (Lane 5). When only the nicking endonuclease was added, its recognition site was not generated (Lane 4), a similar result to that of lane 3. In the presence of OTA, the addition of polymerase and nicking endonuclease resulted in the successful extension of primer and production of ssDNA (line 1), suggesting the performance of SDA and good operation of the designed aptamer.

### 3.3. Optimization of the Experimental Conditions

To obtain optimum sensing performance of OTA detection, several key factors were investigated in the presence of 500 ng mL^−1^ OTA, including the incubation times of HP and OTA; the concentrations of nicking endonuclease, *Bst* DNA polymerase and Mg^2+^; the ratio of primer to HP; and the reaction temperature (Figure 2). The combination degree of OTA and HP is the premise of subsequent amplification and directly affects the final result. Therefore, the incubation time of OTA and HP was first optimized in this study (Figure 2A). With the extension of the incubation time, the amplification efficiency gradually increased and reached a plateau at 30 min. After that, the *Ct* value did not change significantly over time, indicating that the combination of OTA and HP achieved maximum efficiency. Thus, 30 min was selected as the optimal incubation time and used in subsequent experiments. Then, nicking endonuclease with activities ranging from 0.5 U μL^−1^ to 2.5 U μL^−1^ were added to the solution to investigate the optimal endonuclease concentration (Figure 2B). The result showed that *Ct* first decreased and then increased gradually when nicking endonuclease activity was over 1.0 U μL^−1^. When *Ct* reached the lowest value, the amplification efficiency reached the highest. Thus, the optimal activity of endonuclease was 1.0 U μL^−1^. We also analyzed the effect of *Bst* DNA polymerase concentration on the amplification efficiency (Figure 2C). According to the result, 0.8 U μL^−1^ was selected as the reasonable concentration of polymerase in the following experiments. Mg^2+^ is an activator of *Bst* DNA polymerase. If its concentration is too low, the activity of DNA polymerase will be insufficient, while a high concentration of Mg^2+^ will result in non-specific amplification. The results showed that 7 mM was the optimum Mg^2+^ concentration (Figure 2D). The primer specifically recognizes and binds to the HP stem, and their effective combination determines the subsequent amplification efficiency. Excessive primer concentration tends to form primer dimers. Therefore, the ratio of primer to HP (4:5, 9:10, 1:1, 10:9, 6:5) is optimized. As shown in Figure 2E, *Ct* decreased with the increase in the primer-to-HP ratio and reached a plateau when their ratio was 1:1. Thus, the optimal primer to HP ratio was 1:1 with a primer concentration of 500 nM. In addition, the amplification process required the cooperation of two enzymes, and their optimal reaction temperatures were also different. Because of that, the temperature directly affected the enzyme activity, so the temperature was also optimized (Figure 2F). The results showed that these two enzymes had the best synergistic effect and the highest amplification efficiency at 65 °C, and therefore 65 °C was selected for the amplification reaction.

### 3.4. Analytical Performance

Under the optimized conditions, the feasibility of the proposed aptasensor was investigated upon the addition of different concentrations of OTA with real-time fluorescence curves. The concentration of OTA determines the concentration of the opened HP and is also the template concentration involved in the SDA amplification. Different *Ct* values were obtained following the addition of different concentrations of OTA, generating a real-time fluorescence curve (Figure 3). We found that the *Ct* of the curve decreased dramatically with increases in OTA concentration (0.001–100 ng mL^−1^) until it reached the plateau. There was a good linear relationship between *Ct* and the logarithm of OTA concentration in the range of 0.01–50 ng mL^−1^. The linear regression equation was *Ct* = −1.059 lgC_OTA_ + 14.573, R^2^ = 0.9936, and the LOD for OTA was determined as 0.01 ng mL^−1^ (3 S/N). In comparison with other aptamer-based sensors listed in Table 2, this fluorescent aptasensor demonstrated a lower detection limit and better detection range, with ideal rapidity.

### 3.5. Specificity and Reproducibility

The specificity in this study is mainly affected by the aptamer that binds to the OTA specifically. Therefore, several structural analogues of OTA, including OTB, AFB1, ZEN and FB1, were used to evaluate the specificity of the proposed aptasensor (Figure 4A). Although the concentrations of OTB, AFB1, ZEN and FB1 were 100 ng mL^−1^, which was 10 times that of OTA, no significant *Ct* values were obtained in the absence of OTA. Moreover, a mixture of other mycotoxins did not interfere with the specific detection of OTA. Therefore, the proposed method suggested good specificity toward OTA.

In addition, six parallel detections were performed to verify the reproducibility of the aptasensor at concentrations of 0.01 and 100 ng mL^−1^ OTA (Figure 4B). The relative standard deviation values were 1.1% and 1.0%, respectively, indicating the satisfactory reproducibility of the developed sensor.

### 3.6. Real Samples Detection

To estimate the feasibility of the designed aptasensor for OTA detection, wheat and oats samples were spiked with 1 and 10 ng mL^−1^ OTA, respectively, by manual standard addition, and their recovery rates were calculated. Meanwhile, ELISA was used to assist in verifying the detection results of the fluorescent aptasensor. The results indicated that the analytical results detected by the developed aptasensor were consistent with ELISA (Table 3). Moreover, the recovery rate of our method was 96~100.1%, which met the requirements of the European Commission. The results suggested that the proposed aptasensor showed good performance for OTA determination in real sample analysis.

## 4. Conclusions

In conclusion, we developed an SDA-based fluorescent aptasensor for the detection of OTA. The HP structure contained an OTA-specific aptamer that greatly eliminated the influence of other interfering substitutes. The proposed sensor exhibited excellent specificity and sensitivity for the determination of OTA within 60 min and could be successfully applied to wheat and oat samples with recovery of 96~100.1%. In addition, the operation procedure was simple and had good reproducibility. This newly established SDA-based fluorescent aptasensor had multiple advantages, such as simple operation, rapid detection, high sensitivity and good reproducibility; thus, it will be expected to be used in contaminants in food.

## Data Availability

Data is contained within the article.

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
