# Peer review of "A Novel Fluorescent Aptasensor Based on Real-Time Fluorescence and Strand Displacement Amplification for the Detection of Ochratoxin A"

_foods, 2022, doi:10.3390/foods11162443_

Round 1

Reviewer 1 Report

The manuscript « A novel fluorescent aptasensor based on real-time fluorescence  and strand displacement amplification for detection of ochra- toxin A” by Guo et al., presents a fluorescent sensor based on OTA-aptamer which opens upon binding the target OTA and enable amplification with SDA primers. The amplified products are dyed with SYBR Green II and generated fluorescence is detected.

The manuscript is well written and the subject is of high importance.

Introduction provides state-of-the –art regarding traditional and official methods used for OTA monitoring in food but not previously reported biosensors for its detection. A comparative table should be added at the end of results/discussion part to compare previous biosensors with this one in term of sensitivity, rapidity, and other analytical parameters.

Author Response

Reviewer 1:

Comment: The manuscript “A novel fluorescent aptasensor based on real-time fluorescence and strand displacement amplification for detection of ochra- toxin A” by Guo et al., presents a fluorescent sensor based on OTA-aptamer which opens upon binding the target OTA and enable amplification with SDA primers. The amplified products are dyed with SYBR Green II and generated fluorescence is detected.

The manuscript is well written and the subject is of high importance.

Response: Thanks a lot for your appreciation and recommendation of our manuscript. This manuscript has been carefully revised according to the reviewer’s kind suggestions.

Comment: Introduction provides state-of-the –art regarding traditional and official methods used for OTA monitoring in food but not previously reported biosensors for its detection.

Response: Thank you for your kind suggestion. In the section of “Introduction”, we have reported several aptasensors used for OTA detection as follows:

“Taking advantages of those excellent characteristics of aptamers, various aptasensors have been built for ultrasensitive detection of OTA through different sensing platforms so far, such as electrochemical[19;20] and optical[21-23] platforms.”(Page 2, Lines 69-72)

Zhao, Y.; Liu, R.; Sun, W.; Lv, L.; Guo, Z. Ochratoxin A detection platform based on signal amplification by Exonuclease III and fluorescence quenching by gold nano-particles. Sens. Actuator B-Chem. 2018, 255, 1640-1645.

Jiang, L.; Qian, J.; Yang, X.; Yan, Y.; Liu, Q.; Wang, K.; Wang, K. Amplified impedimetric aptasensor based on gold nanoparticles covalently bound graphene sheet for the picomolar detection of ochratoxin A. Anal. Chim. Acta 2014, 806, 128-135.

Hu, Z.; Lustig, W.P.; Zhang, J.; Zheng, C.; Wang, H.; Teat, S.J.; Gong, Q.; Rudd, N.D.; Li, J. Effective detection of mycotoxins by a highly luminescent metal−organic framework. J. Am. Chem. Soc. 2015, 137, 16209-16215.

Zhang, J.; Xia, Y.K.; Chen, M.; Wu, D.Z.; Cai, S.X. Liu, M.M.; He, W.H.; Chen, J.H. A fluorescent aptasensor based on DNA-scaffolded silver nanoclusters coupling with Zn(II)-ion signal-enhancement for simultaneous detection of OTA and AFB(1). Sensor Actuat. B-Chem. 2016, 235, 79-85.

Wang, B.; Wu, Y.; Chen, Y.; Weng, B.; Xu, L.; Li, C. A highly sensitive aptasensor for OTA detection based on hybridization chain reaction and fluorescent perylene probe. Biosen. Bioelectron. 2016, 81, 125-130.

Comment: A comparative table should be added at the end of results/discussion part to compare previous biosensors with this one in term of sensitivity, rapidity, and other analytical parameters.

Response: In the section of “Results”, a comparative table (Table 2) has been added to compare previous biosensors with this one in term of limit of detection (LOD), linear range, and detection time.

Table 2. Comparison of SDA based fluorescent aptasensor with other technologies based on aptamers for OTA detection.

Analysis method

Linear range

LOD

Detection time

Ref

Aptamer/NH2 Janus particles based Electrochemical aptasensor

1×10−5-10 nM

3.3 × 10−3 pM

30 min

[38]

Hydrogel/chitosan based Label-free impedimetric electrochemical sensor

0.1-100 ng mL-1

30 pg mL−1

90 min

[39]

Fluorescent aptasensor by using cascade strand displacement reaction

1-1000 ng mL-1

0.63 ng.L−1

110 min

[40]

Impedimetric aptasensor based on Pencil Graphite Electrodes

0.1-2.0 ng mL-1

0.1 ng.L−1

90 min

[41]

Label-free chemiluminescence biosensor

0.1-2.0 ng mL-1

0.07 ng.L−1

90 min

[42]

Fluorescent aptasensor based on SDA

0.01-50 ng mL-1

0.01 ng mL-1

60 min

This study

Reviewer 2 Report

The authors wish to report a fluorescent aptasensor for detection of ochratoxin A, which is a toxic contaminant presents in food. This new developed sensor is a promising platform of Orchatoxin A (OTA) detection in food sample for its highly sensitive and preciseness of fluorescence detection technology and the simplicity of SDA operation. The recovery rate of this method is able to fulfill the requirements of the European Commission so this aptasensor gives good performance for OTA detection in real food sample. The manuscript needs minor revision to address the specific comment mentioned below.

Specific comments:

1. Please mention in brief about the sensing mechanism in Figure 1 caption.

2. What is the reason of fluorescence enhancement of SYBR Green II in presence of DNA strands? Mention it more clearly.

Author Response

Reviewer 2:

Comments and Suggestions for Authors

The authors wish to report a fluorescent aptasensor for detection of ochratoxin A, which is a toxic contaminant presents in food. This new developed sensor is a promising platform of Orchatoxin A (OTA) detection in food sample for its highly sensitive and preciseness of fluorescence detection technology and the simplicity of SDA operation. The recovery rate of this method is able to fulfill the requirements of the European Commission so this aptasensor gives good performance for OTA detection in real food sample. The manuscript needs minor revision to address the specific comment mentioned below.

Response: Thanks a lot for your appreciation and recommendation of our manuscript. This manuscript has been carefully revised according to your kind suggestions.

Specific comments:

  1. Please mention in brief about the sensing mechanism in Figure 1 caption.

Response: Thank you for your helpful suggestion. The sensing mechanism was added in Figure 1 caption, as follows:

“In the presence of OTA, the random coil ssDNA structure of OTA aptamer is transformed to an antiparallel G-quadruplex, exposing the binding sites in the stem of HP. The primers specifically recognized and bonded to the binding sites to initiate the SDA, thus producing large numbers of repetitive ssDNA. The SYBR Green II binds to ssDNA and intense fluorescence is emitted. Whereas, in the absence of OTA, the stem-loop structure of HP was stable and SDA could not be initiated, resulting in the detachment of SYBR Green II dye from ssDNA and generating a negligible fluorescent signal.” (Page 4, Lines 172-178)

  1. What is the reason of fluorescence enhancement of SYBR Green II in presence of DNA strands? Mention it more clearly.

Response: Thanks a lot for your kind suggestion. We have added the reason of fluorescence enhancement of SYBR Green II in presence of DNA strands in the text as follows:

“When SYBR Green II binds to ssDNA, the energy of the electrons from the ground state to the excited state is equal to that of the excitation of the light photon, so it can absorb the photon. And the electrons are excited to the excited state and then back to the lower state, thus emitting intense fluorescence.”(Page 4, Lines 163-166)

Reviewer 3 Report

The Authors have reported a fluorescence method to detect Ochratoxin A. The research work has been done very well and the discussion section was clear. They explained the preparing steps and measuring prosses very well and that they are not only respect to your self but also respect to readers. They are just minor things that I think they should mention them. The detection limit should have been reported as (3 S/N). Reporting some parameters will increase the quality of the paper and help it be cited in the future. for example, it is better if they report the Langmuir isotherm constant (KL), dissociation constant (Kd), and the maximum number of binding sites (Imax)  and selectivity factor.

congratulations   

Author Response

Reviewer 3:

The Authors have reported a fluorescence method to detect Ochratoxin A. The research work has been done very well and the discussion section was clear. They explained the preparing steps and measuring prosses very well and that they are not only respect to your self but also respect to readers. They are just minor things that I think they should mention them. The detection limit should have been reported as (3 S/N). Reporting some parameters will increase the quality of the paper and help it be cited in the future. for example, it is better if they report the Langmuir isotherm constant (KL), dissociation constant (Kd), and the maximum number of binding sites (Imax)  and selectivity factor.

Response: Thanks a lot for your appreciation and recommendation of our manuscript. This manuscript has been carefully revised according to your kind suggestions.

(1) The detection limit has been reported as 3 S/N, as follows: “And the LOD for OTA was determined as 0.01 ng mL-1(3 S/N).” (Page 7, Lines 254-255)

(2) In this study, the OTA specific aptamer sequence was designed based on (5′-GATCGGGTGTGGGTGGCGTAAAGGGAGCATCGACA-3′) according to Cruz et al. (2008). We have added the Kd value as follows: “OTA specific aptamer sequence was designed based on (5′-GATCGGGTGTGGGTGGCGTAAAGGGAGCATCGACA-3′), with dissociation constant (Kd) of 0.2 μmol/L, according to Cruz et al. (2008)[36].”(Page 3,Lines 107-110). The aim of this study was to establish an ultrasensitive and efficient fluorescent aptasensor based on SDA, which can be used to determinate real food samples with excellent analysis performance. So, we now focus on the study of its feasibility, detection limit, experimental conditions, specificity, reproducibility and real sample analysis. In subsequent study, we will conduct some experiments to evaluate the binding affinity, Langmuir isotherm constant (KL), dissociation constant (Kd), and the maximum number of binding sites (Imax) for OTA-aptamer with OTA.

Cruz-Aguado, Jorge A.; Penner, Gregory. Determination of Ochratoxin A with a DNA Aptamer. Journal of Agricultural and Food Chemistry 2008, 56(22), 10456-10461.
